# Orbital Radiotherapy for Graves’ Ophthalmopathy: Single Institutional Experience of Efficacy and Safety

**DOI:** 10.3390/diseases13020061

**Published:** 2025-02-17

**Authors:** Madalina La Rocca, Barbara Francesca Leonardi, Maria Chiara Lo Greco, Giorgia Marano, Roberto Milazzotto, Rocco Luca Emanuele Liardo, Grazia Acquaviva, Viviana Anna La Monaca, Vincenzo Salamone, Antonio Basile, Pietro Valerio Foti, Stefano Palmucci, Emanuele David, Silvana Parisi, Antonio Pontoriero, Stefano Pergolizzi, Corrado Spatola

**Affiliations:** 1Radiation Oncology Unit, Department of Biomedical, Dental and Morphological and Functional Imaging Sciences, University of Messina, 98122 Messina, Italy; barbarafrancesca.leonardi@studenti.unime.it (B.F.L.); mariachiara.logreco@studenti.unime.it (M.C.L.G.); giorgia.marano@studenti.unime.it (G.M.); silvana.parisi@unime.it (S.P.); apontoriero@unime.it (A.P.); stefano.pergolizzi@unime.it (S.P.); 2Radiation Oncology Unit, Department of Medical Surgical Sciences and Advanced Technologies “G.F. Ingrassia”, University of Catania, 95123 Catania, Italy; r.milazzotto@policlinico.unict.it (R.M.); l.liardo@policlinico.unict.it (R.L.E.L.); gra.acquaviva@libero.it (G.A.); v.lamonaca@policlinico.unict.it (V.A.L.M.); v.salamone@policlinico.unict.it (V.S.); 3Department of Medical Surgical Sciences and Advanced Technologies “G.F. Ingrassia”, University Hospital Policlinico “G. Rodolico-San Marco”, 95123 Catania, Italy; a.basile@unict.it (A.B.); p.foti@policlinico.unict.it (P.V.F.); spalmucci@unict.it (S.P.); 4Radiology I Unit, Department of Medical Surgical Sciences and Advanced Technologies “G.F. Ingrassia”, University of Catania, 95123 Catania, Italy; e.david@unict.it

**Keywords:** Graves’ ophthalmopathy, thyroid eye disease, radiotherapy, exophthalmos, Graves’ orbitopathy, orbital irradiation, combined treatment

## Abstract

Graves’ ophthalmopathy is the most common extrathyroidal manifestation of Graves–Basedow disease. Radiotherapy is effective especially when used in synergy with the administration of glucocorticoids. The aim of our study was to analyze the effectiveness and safety of radiotherapy, using different protocols, to improve ocular symptoms and quality of life. Methods: We retrospectively analyzed the clinical data of two-hundred and three patients treated with retrobulbar radiotherapy between January 2002 and June 2023. Ninety-nine patients were treated with a schedule of 10 Gy in 10 fractions and one-hundred and four were treated with 10 Gy in 5 fractions. Radiotherapy (RT) was administrated during the 12 weeks of pulse steroid therapy. Patients were evaluated with a clinical exam, orbital CT, thyroid assessment, and Clinical Activity Score (CAS). Results: The median follow-up was 28.6 months (range 12–240). Complete response was found in ninety-four pts (46.31%), partial response or stabilization in one hundred pts (49.26%), and progression in nine pts (4.43%). In most subjects, an improvement in visual acuity and a reduction in CAS of at least 2 points and proptosis by more than 3 mm were observed. Three patients needed decompressive surgery after treatment. Only G1 and G2 acute eye disorders and no cases of xerophthalmia or cataract were assessed. Conclusions: RT is an effective and well-tolerated treatment in this setting, especially when associated with the administration of glucocorticoids. Although the most used fractionation schedule in the literature is 20 Gy in 10 fractions, in our clinical practice, we have achieved comparable results with 10 Gy in 5 or 10 fractions with a lower incidence of toxicity.

## 1. Introduction

Graves’ ophthalmopathy (GO) is an inflammatory pathology of the orbit associated with an underlying autoimmune pathogenesis [1,2]. GO represents the most frequent extrathyroidal manifestation and its associated with hyperthyroidism in 90% of cases (primarily with Graves’ disease, less frequently with Hashimoto’s thyroiditis, myxedema without previous thyrotoxicosis) [3]. GO primarily involves inflammatory changes in orbital tissues, especially retrobulbar soft tissues with potential thickening and fibrosis of the extraocular muscles (EOMs) and orbital fat, increasing the volume within bony orbit [4]. The principal cause is the production of antibodies directed against thyroid stimulating hormone (TSH) receptors located in the retro-orbital space, increasing the productions of various cytokines [5,6]. These cytokines stimulate fibroblasts to produce glycosaminoglycan that, when hydrophilic, results in edema, causing marked swelling of the orbital tissue in GO [1]. The symptomatic presentations of GO are a direct result of the inflammatory and fibrotic reactions in the retro-orbital space [5]. Incidence is higher between the second and sixth decade of age, with a peak between 40 and 60 years of age, especially in women (female/male ratio of 5:1). It can be predominantly bilateral, but unilateral and asymmetrical forms are also documented [7]. GO is accompanied by complex ocular symptoms that can be debilitating and impair the quality of life (QoL) of the affected individual, with effects on work and social relationships [8]. The clinical evolution of GO is characterized by a progressive deterioration that can be divided into three phases: active phase or florid, characterized by increased activity, where symptoms and signs worsen rapidly, reaching a point of maximal severity, followed by a static plateau period, with a gradual improvement toward the baseline, and finally a phase of progressive shutdown or inactivation [9]. This cycle has a variable duration from individual to individual, but usually it does not exceed 12–24 months [10]. To ensure stable and lasting control of Graves’ ophthalmopathy, it is crucial to implement behavioral strategies and, when necessary, pharmacological therapies from the outset. This includes maintaining a euthyroid state, managing metabolic syndrome, diabetes, hypercholesterolemia, and quitting smoking [11,12,13,14].

Treatment options are numerous and can be tailored to patients depending on the severity and time from the onset of the disease. Among them, radiotherapy has been used for decades for ocular pathologies with optimal results [15,16,17,18,19].

The aim of our study was to retrospectively compare the effectiveness in symptoms control and safety between two populations of patients treated with intravenous steroid therapy and subjected to radiotherapy with the same total dose but two different fractionation schedules.

## 2. Materials and Methods

Between January 2002 and June 2023, two-hundred and three patients with GO diagnosis were treated with retrobulbar radiotherapy in the Oncologic Radiotherapy ward of “G. Rodolico” Hospital in Catania. We retrospectively evaluated the clinical reports of these patients, using the inclusion and exclusion criteria, and the study design is represented in Table 1 and Figure 1, respectively.

Patients were assessed at the baseline with a clinical examination comprising an ocular exam with visual field, orbital computer tomography (CT), thyroid function assessment (including thyroid stimulating hormone TSH, T3, T4, free thyroxine FT4 levels), CAS (Clinical Activity Score) (Table 2), clinical assessment of GO severity (Table 3), and proptosis measured with the Hertel ophthalmometer [14].

A CT scan without contrast with a slice thickness of 2 mm was performed for image acquisition and target contouring. Patients were immobilized with a custom-made thermoplastic mask in supine position. We also used a device called “Eye Bridge” (Figure 2), endowed with a central red laser for sight fixation during the CT and the subsequent fractions that helped us to reduce the dose to both lenses.

The treatment was planned on MOSAIQ^®^ (IMPAC Medical Systems, Sunnyvale, CA, USA) for patients treated before 2020 and RayStation^®^ (RaySearch Medical Laboratories AB, Stockholm, Sweden) for the rest of the patients; we provide an example of this plan in Figure 3.

The clinical target volume (CTV) was defined as equivalent to the PTV (planning target volume) and included retro-orbital fat and orbital muscles. Bilateral lenses, eyeballs, optic nerves, lacrimal glands, chiasm, brain, and brainstem were contoured as organs at risk (OARs). Before every treatment session, a portal image was taken daily to help minimize set-up uncertainties, making the treatment more reproducible, accurate, and precise.

All patients were treated with Three-Dimensional Conformational Radiation Therapy (3DCRT) using a 6 MV photon beam linear accelerator (with a linear accelerator-based Oncor^TM^, Siemens Medical Solutions, Erlangen, Germany), administrated with 2 lateral fields, tilting the beams posteriorly with a personalized angle between 5 and 7 degrees to spare the lens and contralateral ocular areas.

Radiotherapy (RT) was imbricated during the 12 weeks of pulse steroid therapy (weekly hydrocortisone i.v. bolus).

Patients underwent the first clinical revaluation at the end of the RT course by the radiation oncologists. The follow-up was continued at one month and then three months for the first two years, and then every year. During the follow-up, all patients underwent separate evaluations by the ophthalmologist and endocrinologist. Depending on the specialists’ clinical practice, orbital CT scans were performed as needed. Within the six months of follow-up, all patients had an ophthalmologic assessment, including ophtalmometric proptosis evaluation.

The response to orbital RT was defined as an improvement in CAS of at least two points, a reduction in proptosis by at least 2 mm, an improvement in diplopia and edema, a reduction in lacrimation and pain and pain during eye movements, and an improvement in visual acuity.

Treatment-related toxicities were graded according to the Common Terminology Criteria for Adverse Events (version 5.0) [20]. Acute toxicities were defined as adverse events occurred during or within 90 days from the end of treatment and late toxicity as adverse events occurred from 90 days after the end of radiotherapy.

## 3. Results

Among two-hundred and three patients, 66.6% and 65.38% were females patients treated with 10 fractions and 5 fractions, respectively. The two cohorts displayed similar characteristics in terms of age, gender, smoking habits, diabetes, serum parameters, symptom onset timing, disease activity, and severity based on the patient profiles rather than statistical evaluation. Ninety-nine patients were treated using a schedule of 10 Gy administrated in 10 fractions and, due to the concomitant COVID-19 pandemic, the other one-hundred and four patients were treated with a shortened and hypofractionated schedule of 10 Gy in 5 fractions. All the population characteristics at baseline, the CAS, and symptoms are summarized in Table 4.

The median follow-up was 28.6 months (range 12–240). Complete response was found in ninety-four patients (46.31%), partial response or stabilization in one hundred patients (49.26%), and progression in nine patients (4.43%). All patients completed their treatment on time without any interruptions. During the first six months of follow-up, an improvement in visual acuity and a reduction in CAS of at least 2 points and proptosis by more than 3 mm were observed in 86.4% and 74.6% of patients, respectively. Only three patients needed decompressive surgery after treatment. There were only 30 cases of G1 acute blurred vision, 80 cases of G1 dry eye treated with lubricants, and 5 cases of G2 eye pain. In all these cases, the symptoms resolved spontaneously, without leaving long-term sequelae. No cases of xerophthalmia or cataracts were assessed. The changes compared to the initial symptoms and their progression are outlined in Table 5 and Table 6 and are represented graphically in Figure 4. At the 9- and 12-month follow-ups, none of the patients in either study cohort reported tearing or eye burning.

## 4. Discussion

Radiotherapy (RT) has been used in thyroid-associated orbitopathy (TAO) since 1913 [21]. It is known that ionizing radiation at low doses of 0.1–1 Gy functionally modulates inflammatory cells, especially lymphocytes, and activates heat shock proteins; the induction of nitric oxide synthase in activated macrophages may also be involved in the anti-inflammatory response to radiation. While the anti-inflammatory effect of retrobulbar irradiation is evident at low doses such as 2.4 Gy, higher doses are necessary to avoid the synthesis of glycosaminoglycans by orbital fibroblasts [15,22,23]. On the other hand, higher doses may have a negative impact by inducing the fibrosis of the extraocular muscles [24]. Although retrobulbar RT for GO has been routinely used for decades, data on the optimal dose and fractionation schedules to be administered to obtain the maximum response in the absence of complications are still controversial. From the 1970s to present day, several studies have been carried out that aimed to evaluate the effectiveness of retrobulbar radiotherapy in GO, documenting also a few side effects [5,15,16,17,25,26,27,28,29,30,31,32,33,34,35,36]. The best responses were observed on soft tissues, corneal involvement, and a reduction in vision loss, though there were less positive responses in terms of ocular motility and proptosis [37,38,39]. The response to radiotherapy can be negatively influenced by several factors: duration of symptoms (>12 months), male gender, coexistent hyperthyroidism, advanced age, diabetes, and cigarette smoking, while the female sex constitutes to be a favoring factor [11,40,41]. Radiotherapy should be recommended in the early stages of the disease, preferably less than twelve months after the onset of symptoms. The efficacy of orbital radiotherapy can be increased by the synergistic interaction with glucocorticoids, as demonstrated by several studies and metanalyses [16,30,31,42]. It is particularly useful also in treating those patients who are insensitive or tolerant to GC therapy, or who have relapsing symptoms after glucocorticoid therapy [5]. The effects of RT are slower to show compared to GC, but it can be effective for a longer period. Lateral opposing fields (LOFs), three-dimensional conformational radiotherapy (3D-CRT), and intensity-modulated radiation therapy (IMRT) are all alternative techniques [33,43,44]. We assessed complete response in 46.31% and stabilization in 49.26% without having to resort to further radiation or corticosteroid retreatments, regardless of the presence of risk factors that could have had a negative impact. The most used treatment scheme is 20 Gy administrated in 10 fractions, but different doses and schedules were investigated, as summarized in Table 7 [5,8,17,38,41,45].

Nakahara et al. [46] evaluated two groups of patients treated with 10 Gy (15 patients) and 24 Gy (16 patients) in association with glucocorticoids (GCs). After a follow-up of 1–3 months, the authors concluded that a dose of 24 Gy is more effective than 10 Gy. Choi et al. [15] used 24 Gy in 12 fractions; Gerling et al. [49] compared total doses of 2.4 Gy and 16 Gy, with no significant differences shown in either group. They concluded that retrobulbar irradiation for ophthalmopathy should not exceed 2.4 Gy. Weissmann et al. [48] confronted two schedules: a low dose of 4.8 Gy in fractions of 0.8 Gy and a high dose of 20 Gy in 10 fractions with no significant difference in the overall improvement in symptoms, but they found that patients treated with lower doses of radiation needed a second series of radiotherapy significantly more frequently than patients treated with high-dose RT. Johnson et al. [47] performed a retrospective study on 129 patients who underwent doses of 12, 16, or 20 Gy. They reported that 12 Gy may be more effective in patients who suffer from dysmotility. Kahaly et al. [37] demonstrated that the same efficacy is obtained with 10 Gy in 10 fractions. In the recent EUGOGO guidelines, both 10 Gy and 20 Gy are considered acceptable alternatives [14]. In Ohtsuka et al.’s study [4], orbital RT after corticosteroid pulse therapy was not associated with beneficial therapeutic effects on rectus muscle hypertrophy or the proptosis of active GO during the 6-month follow-up period. However, in Gorman et al.’s study [34], a prospective randomized trial, forty-two patients were treated with 20 Gy of EBRT and sham therapy on the other side with the reversion of therapy six months later; no clinical or statistical difference was observed.

In our experience of comparing a dose of 10 Gy in ten or five fractions, we did not find a statistically significant difference in the outcome measures after 1, 3, 6, or 12 months, but we did find a slower but steady improvement after longer follow-ups. We also found a better patient-related perception of shorter treatments. Depending on the severity of the symptoms and the time elapsed since their onset, radiotherapy or glucocorticoids associated with it may not be sufficient. New molecules are becoming more relevant in the treatment of patients with moderate or severe GO. Rituximab (a chimeric human and mouse MAB against CD20) can be used at 100 mg or 500 mg dosages [50,51]. An American study [52] confronted rituximab versus placebo, where it showed no difference in reducing CAS or the severity of GO in both groups. In contrast, an Italian study [53] demonstrated better ophthalmic and QoL outcomes with rituximab as compared to i.v. GC administration. Tocilizumab is a humanized MAB against the interleukin (IL)-6 receptor approved for use in rheumatoid arthritis. Data suggest that tocilizumab may cause a rapid resolution of inflammatory signs with benefits predominantly on soft tissue; it is generally well tolerated but with a higher rate of infections and headache [54]. Teprotumumab, a fully human monoclonal insulin-like growth factor-I receptor (IGF-IR) inhibitory antibody (administrated in eight infusions: 10 mg/kg for the first infusion, 20 mg/kg for subsequent infusions over a total of 21 weeks), showed improvements in GO signs and symptoms, with the long-term maintenance of responses [1,37,55]. The safety and efficacy of teprotumumab were evaluated in two trials: OPTIC [23] and OPTIC-X [56], and its incorporation into routine clinical practice is currently limited by the lack of comprehensive long-term efficacy and safety data, the absence of a head-to-head comparison with i.v. glucocorticoids, restricted geographical availability, and costs. The American Thyroid Association/European Thyroid Association Consensus Statement recommended the first-line use of teprotumumab for patients with moderate–severe disease presenting with proptosis and/or diplopia [14]. In clinical practice, however, due to the advent of systemic therapies such as monoclonal antibodies, the role of radiotherapy is increasingly reduced and usually reserved for patients with multiple comorbidities, with relative or absolute contraindications, or those who refuse these treatments.

This study has several limitations due to its retrospective design. Selection bias may have occurred, as treatment decisions were influenced by physician preferences and patient characteristics. Despite including all eligible patients over the 20-year period, unmeasured confounders may still affect the results. Additionally, information bias is a concern, as data were extracted from medical records, and missing or incomplete information could not be fully excluded. Changes in clinical practices over time may have also introduced variability in treatment regimens and supportive care. Another limitation was that the evaluation of proptosis through diagnostic tests was performed at the discretion of the ophthalmologist.

## 5. Conclusions

Despite the increasing role of immunotherapy and different systemic therapeutic strategies in the treatment of patients with moderate–severe GO, the role of radiotherapy as an effective and well-tolerated therapy is still important. Radiotherapy may help in the long-lasting relief of symptoms and improvements in CAS, proptosis, visual acuity, and ocular movement limitations. As reported in the literature, the synergy between radiotherapy and intravenous steroid therapy leads to excellent results. Although the most commonly used fractionation schedule is 20 Gy in 10 fractions, our study has shown comparable outcomes with lower-dose regimens, such as 10 Gy in 5 or 10 fractions, which also demonstrate equivalent control while resulting in fewer toxic effects.

## Figures and Tables

**Figure 1 diseases-13-00061-f001:**
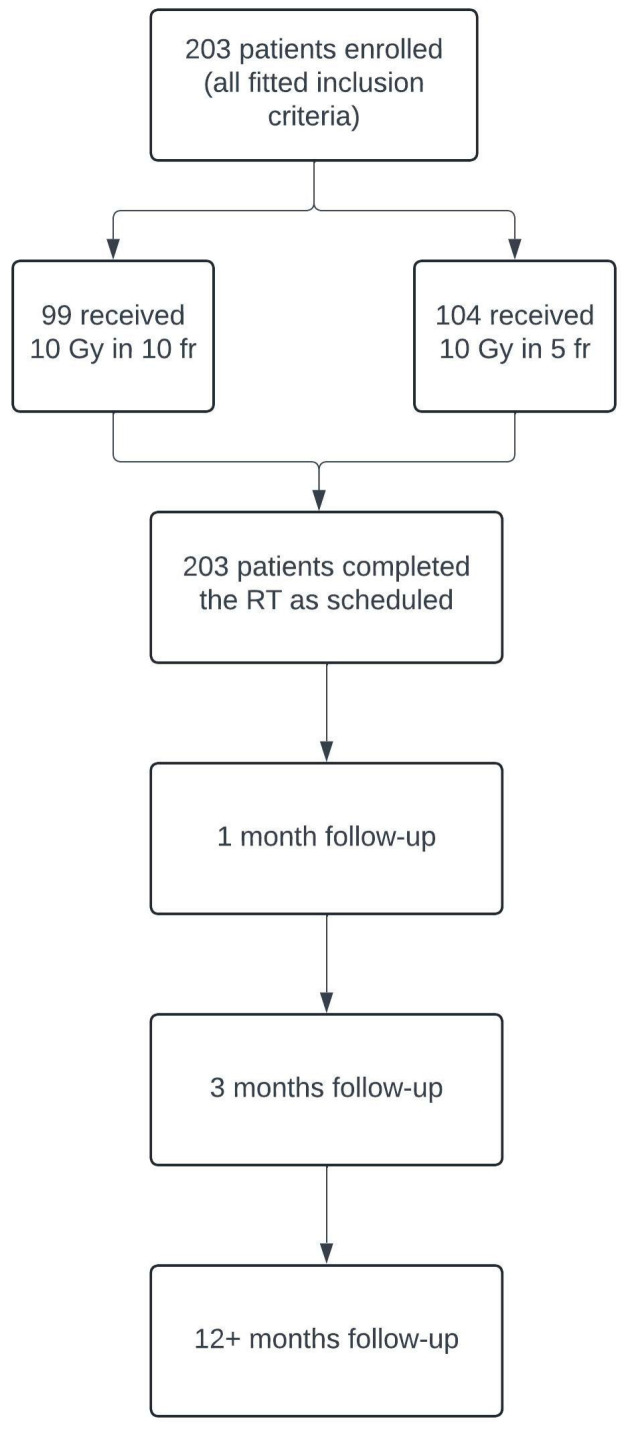
Flowchart illustrating the design of our retrospective study, including patient enrollment and follow-up process.

**Figure 2 diseases-13-00061-f002:**
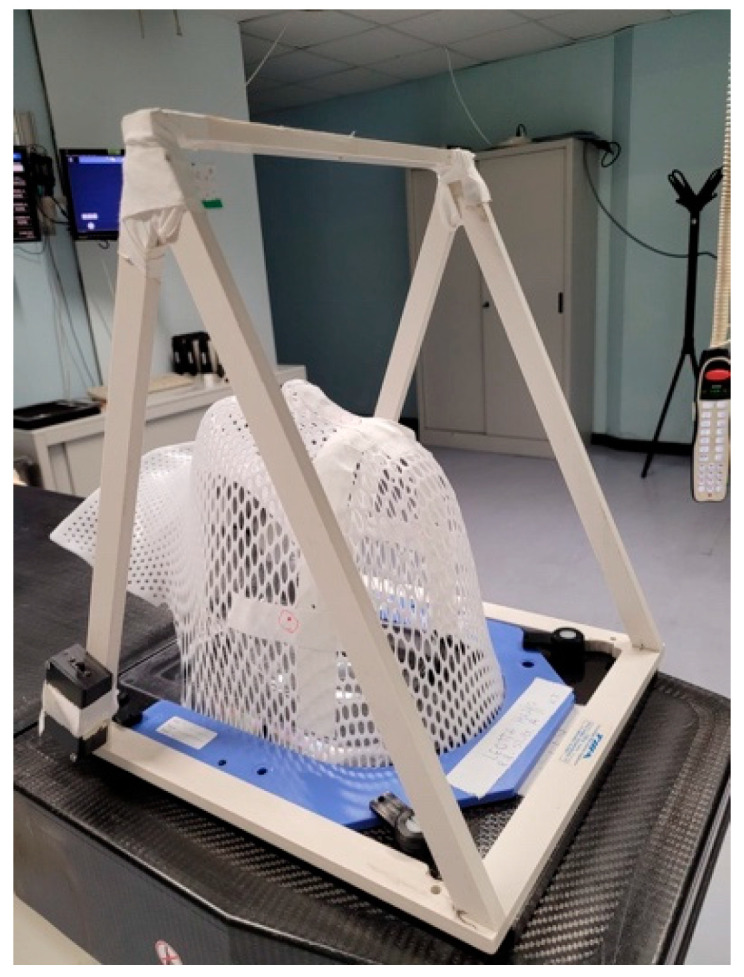
The immobilization device in this image is called “Eye Bridge” or “Eye Fixation System”. It is a plastic structure fitted to frame the thermoplastic base. A red laser is located in the central horizontal bar positioned like a bridge above the orbits. During the CT simulation and treatment phases, patients are instructed to keep their gaze fixed on the red light to better preserve the crystalline lenses.

**Figure 3 diseases-13-00061-f003:**
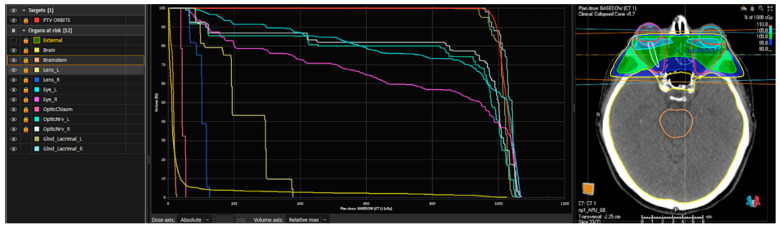
An example of a dose–volume histogram (DVH) and a plan rendering of a patient treated with 10 Gy in 5 fractions performed using Raystation^®^. From left to right: the planning target volume (PTV) in red includes the retro-orbital space; the other indicated structures correspond to the organs at risk (OARs) (external referring to the patient’s skin, brain and brainstem, optic chiasm, optic nerves, lenses, eyeballs, and lacrimal glands on both sides). In the central section, the DVH (dose–volume histogram) is shown, graphically representing the dose distribution to the PTV (in red) and the organs at risk. On the right side, an example of a treatment plan is displayed, including the PTV, OARs, and the treatment field coverage.

**Figure 4 diseases-13-00061-f004:**
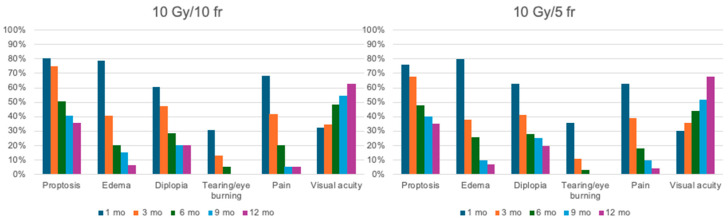
Graphic representation of outcome. On the left is the treatment response in patients treated with 10 Gy in 10 fractions; on the right is the response in patients treated with 10 Gy in 5 fractions. Patients from both treatment arms were evaluated at 1, 3, 6, 9, and 12 months after the completion of radiotherapy.

**Table 1 diseases-13-00061-t001:** Inclusion and exclusion criteria.

Inclusion Criteria:	Exclusion Criteria:
Age > 18 years. Active GO with CAS > 3 or CAS = 3 with severely progressive dysmotility. CT of the orbits shows characteristic features of GO (thickening of straight muscles in both orbits, increased fat tissue, damaged optic nerve, enlargement of the lacrimal gland). Administration of glucocorticoids i.v. Duration of onset symptoms (≤12 months). Start of combined treatment within the first year after initial diagnosis of GO. Bilateral disease. Follow-up of at least 6 months following irradiation.	Inactive disease. More than 1 year after the initial diagnosis. Optic nerve compression orbital decompression surgery before RT. Lack of steroid treatment within the past 3 months. More than one course of systemic steroids. Unilateral disease. Previous RT of the eye region. History of other eye diseases (i.e., glaucoma, diabetic retinopathy, maculopathy).

Abbreviations: CAS: clinical activity score; CT: computer tomography; i.v. intravenous; GO: Graves’ ophthalmopathy; RT: radiotherapy.

**Table 2 diseases-13-00061-t002:** Assessment of Graves’s orbitopathy activity [14].

Clinical Activity Score (CAS)
Spontaneous retrobulbar pain Pain on attempted upward or downward gaze Redness of eyelids Redness of the conjunctiva Swelling of the caruncle or plica Swelling of the eyelids Swelling of conjunctiva (chemosis)

**Table 3 diseases-13-00061-t003:** Classification of GO’s severity [14].

Grade	Characteristics
Mild	Minor impact on daily QoL (immunomodulation or surgical treatment are not necessary). One or more of the following: · Minor lid retraction (<2 mm); · Mild soft tissue involvement; · Exophthalmos; · <3 mm above normal for race and gender; · No or intermittent diplopia; · Corneal exposure responsive to lubricants.
Moderate to severe	Not sight-threatening but a sufficient impact on QoL if immunosuppression (if active GO) or surgical intervention (if inactive GO) is needed. Two or more of the following: · Lid retraction ≥ 2 mm; · Moderate or severe soft tissue involvement; · Exophthalmos ≥ 3 mm above normal for race and gender. Inconstant or constant diplopia
Very severe (sight-threatening)	Presence of dysthyroid optic neuropathy and/or corneal breakdown

**Table 4 diseases-13-00061-t004:** Baseline patient characteristics (n = 203).

Total (n = 203)	10 Gy/10 fr (n = 99)	10 Gy/5 fr (n = 104)
**Median Age**	56.4 (range: 36–78)	
**Sex**		
Male	33 (33.3%)	36 (34.6%)
Female	66 (66.6%)	68 (65.38%)
**Smoking**		
Yes	31 (31.3%)	28 (26.9%)
No	68 (68.7%)	76 (73.1%)
**Diabetes**		
Yes	12 (12.1%)	18 (17.3%)
No	87 (87.9%)	86 (82.7%)
**Thyroid status at diagnosis**		
Normal	64 (64.6%)	72 (30.8%)
Hyperthyreosis	35 (35.4%)	32 (69.2%)
**Active treatment of hyperthyreosis**		
Yes	17 (17.2%)	26 (25%)
No	82 (82.8%)	78 (75%)
**Pre-treatment TSH, T3, T4, FT4 levels**		
Normal	90 (91%)	92 (88.5%)
Elevated	9 (9%)	12 (11.5%)
**Interval from onset of ocular symptoms to RT, months** 8.2 (6–11.5)		
**CAS at baseline**		
4	37 (18.2%)	
5	82 (40.3%)	
6	69 (33.9%)	
7	15 (7.4%)	
**Proptosis at diagnosis**		
**3 mm**	28 (13.79%)	
**4 mm**	115 (56.65%)	
**5 mm**	60 (29.56%)	
**Symptoms at diagnosis**		
Proptosis	120 (63.8%)	
Edema	180 (95.7%)	
Diplopia	125 (66.5%)	
Tearing/eye burning	86 (44.7%)	
Pain	74 (39.4%)	
Visual acuity	103 (54.8%)	

**Table 5 diseases-13-00061-t005:** Symptoms, proptosis, and CAS showed at every follow-up in patients treated with 10 Gy/10 fractions.

10 Gy/10 Fractions	1 Month	3 Months	6 Months	9 Months	12 Months
Proptosis	80%	74.6%	50%	40%	35%
Edema	78%	40%	20%	15%	6%
Diplopia	60%	47%	28%	20%	20%
Tearing/Eye Burning	30%	12.5%	5%	0%	0%
Pain	68%	41%	20%	5%	5%
Visual acuity	32%	34%	48%	54%	62%

**Table 6 diseases-13-00061-t006:** Symptoms showed at every follow-up for the first year after RT in patients treated with 10 Gy/5 fractions.

10 Gy/5 Fractions	1 Month	3 Months	6 Months	9 Months	12 Months
Proptosis	76%	68%	48%	40%	35%
Edema	80%	38%	26%	10%	7%
Diplopia	63%	41%	28%	25%	20%
Tearing/Eye Burning	36%	11%	3%	0%	0%
Pain	63%	39%	18%	10%	4%
Visual acuity	30%	36%	44%	52%	68%

**Table 7 diseases-13-00061-t007:** Studies confronting a different schedule of RT treatment.

Study	N of pts	Dose/Fraction	Technique	Energy	Corticosteroid	Complete Response	Partial Response	Stabilization	Progression/ Recurrence
Nakahara et al., [46] 1995	31	15 pts 10 Gy 16 pts 24 Gy	LOF	4 MeV	3 days iv of 500 mg methylprednisolone. 1 week of os 80 mg prednisolone; tapered until 10 mg/day	24 Gy 9 pts 10 Gy 0	24 Gy 5 10 Gy 10	24 Gy 2 10 Gy 5	24 Gy 4 10 Gy 5
Kahaly et al., [35] 2000	62	A: (18 pts) 20 Gy in 1 Gy/w for 20 weeks B: (22 pts) 10 Gy/10 fr C: (22) 20 Gy/10 fr	LOF	4 MeV	/	A 12 (67%) B 13 (59%) C 12 (55%)	/	/	A 6 (33%) B 9 (41%) C 10 (45%)
Zygulska et al., [45] 2009	121	20 Gy/10 fr	LOF	6 MV	Solu Medrol 2 g/week for 4 weeks.	97 patients (80.2%)	/	21 pts (17.3%)	3 pts (2.5%)
Johnson et al., [47] 2009	129	44 12 Gy 712 fr, 47 16 Gy/8 fr 38 20 Gy/10 fr	LOF	6 MeV	1.5 g 6 weeks (3 weeks 250 mg of prednisolone once/week i.v,			12 Gy 85%, 16 Gy 70% 20 Gy 79%	1 pts with 16 Gy
Matthiesen et al., [5] 2012	209	20 Gy/10 fr	LOF	6 MV	43 pts (20.6%)	93 pts (44.5%)	114 pts (54.5%)	1 pts	1 pts
Weissmann et al., [48] 2020	127	61 4.8 Gy/6 fr 60 20.0 Gy/10 fr	3DCRT	6-MV photon fields.	/	7.9%	55.9%	36.2%	

## Data Availability

The data are not publicly available due to privacy restrictions.

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
