# Peer review of "Orbital Radiotherapy for Graves’ Ophthalmopathy: Single Institutional Experience of Efficacy and Safety"

_diseases, 2025, doi:10.3390/diseases13020061_

Round 1

Reviewer 1 Report

Comments and Suggestions for Authors

This study analyzed the effectiveness and safety of radiotherapy, using different protocols, in improving ocular symptoms and quality of life. However, there are still issues needed to be clarified:

1.     The introduction of some content is relatively small, please enrich the research background.

2.     The contribution of this paper should be written more concisely in the introduction.

3.     The structure of the article should be explained in the introduction section.

4.     Please improve the clarity of the picture.

5.     In Tables 5 and 6, why is Tearing/Eye Burning not recorded at 9 and 12 months?

6.     Please revise the reference formatting to comply with the journal requirements, ensuring that you include recent references from past two years.

Author Response

Comment 1: The introduction of some content is relatively small, please enrich the research background.

Response 1: Thank you for your revision, We have expanded the content of the introduction, paying particular attention to the background of our research and the rationale behind the study.

Comment 2:  The contribution of this paper should be written more concisely in the introduction.

Response 2: Thank you for pointing this out. We have made the section concerning the contribution of our article more concise.

Comment 3: The structure of the article should be explained in the introduction section.

Response 3: We have accordingly included an explanation of the structure of the article in the introduction.

Comment 4: Please improve the clarity of the picture.

Response 4: We have improved the captions to make the figures more understandable for the readers.

Comment 5: In Tables 5 and 6, why is Tearing/Eye Burning not recorded at 9 and 12 months?

Response 5: Thank for you comment. On Tables 5 and 6 we did not report data on tearing/burning eye symptoms, as none of our patients at the 9- and 12-month follow-ups reported or exhibited these symptoms. We also added on the “Result” paragraph on line 176-178 a sentence to clarify it.

Comment 6: Please revise the reference formatting to comply with the journal requirements, ensuring that you include recent references from past two years.

Response 6: We agreeded, and we checked the reference formatting and as suggested we have also revised the references and added more recent bibliographic sources.

Reviewer 2 Report

Comments and Suggestions for Authors

This article demonstrates a well-designed study with comprehensive data spanning 20 years and 203 patients, making it both robust and clinically relevant. The comparison of different radiotherapy fractionation schedules provides valuable insights, and the results are clearly presented with a strong focus on patient-centered conclusions. To improve the manuscript further, consider the following minor revisions: 1. Include statistical metrics, such as p-values or confidence intervals, to strengthen the comparison results; 2. Address study limitations, such as potential biases inherent in the retrospective design; 3. Enhance clarity by correcting minor language errors (e.g., spelling in Figure 2) and providing more detailed figure descriptions to improve professionalism and readability.

Author Response

Comment 1. Include statistical metrics, such as p-values or confidence intervals, to strengthen the comparison results;

Response 1: We thank you for appreciating the effort we put into organizing this study. We conducted a retrospective observational study, focusing on a descriptive analysis of clinical data and the symptomatic benefits observed in patients treated with two different radiotherapy regimens. Given the study's design and primary objective, we have not yet performed formal statistical analyses, such as hypothesis testing with p-values or confidence interval estimations. However, our preliminary findings provide valuable insights into the clinical trends associated with these treatment approaches. Future investigations will incorporate a more detailed assessment of potential prognostic factors that may influence treatment response, alongside robust statistical methodologies to strengthen our conclusions. These analyses will be presented in subsequent publications.

Comment 2. Address study limitations, such as potential biases inherent in the retrospective design;

Response 2: We agree with your comment, We have included the limitations of the study regarding its retrospective design, which can be found at the end of the "Discussion" chapter (line 274-282).

Comment 3. Enhance clarity by correcting minor language errors (e.g., spelling in Figure 2) and providing more detailed figure descriptions to improve professionalism and readability.

Response 3: Thank you for pointing this out. We have corrected typographical errors and improved the figure descriptions to make them more accessible to the readers.

Reviewer 3 Report

Comments and Suggestions for Authors

The authors report their experience with orbital radiotherapy in active TED combined with IV steroids. It is an interesting paper because radiotherapy is debated and more and more marginal with new targeted therapies.

The study has been written by radiotherapists colleagues. The paper lacks of several ophthalmological findings.

Several modificationw are warranted :

- precise the succes: reduction of CAS and proptosis ; reduction of CAS and proptosis ? Please be precise. In the current litterature, success is based on CAS reduction or proptosis reduction by 2 mm and sometimes by composite criteria such as CAS decrease and protposis regression. Response to orbital radiothéerapy should be more explicite.

The proptosis is never measured. How can you say that proptosis has decreased by 3 mm ? By Hertel measurements ? Comparative CT scan ?

A randomized study conducted by the MAYO clinic had shown that radiotherapy was ineffective. It is very frequent study cited. Could you precise the limitations of this study (disease duration, smoking status etc...). In brief, precise the limitations of the inclusion  criteria in randomized study

A maljor limitation is of course the retrospective nature of the study and the lack of control group (only steroids treated patients). This should be clearly highlighted in the limitations of the study.

P value are lacking in table 4. It is said that the groups were comparable but it is not enough for me. The p value would be valuable. The patients are not dichotomized based on the CAS in the table. Why ? Statistics are not provided in the material section.

Table 5 and table 6 : who conducted these results. The ophthalkmology team ? The radiotherapy team ? How was assessed the criteria?

In the conclusion, the authors state that : "Better response rate is observed in case of association with administration of systemic glucocorticoids"; this has not been demonstrated by the study. This study only showed that 10 Gy in 10 or 5 fractions were equally effective and safe.

Author Response

Comment 1 precise the succes: reduction of CAS and proptosis ; reduction of CAS and proptosis ? Please be precise. In the current litterature, success is based on CAS reduction or proptosis reduction by 2 mm and sometimes by composite criteria such as CAS decrease and proptosis regression. Response to orbital radiotherapy should be more explicite.

Response 1: Thank you for your review, which allows us to enrich the article with additional information. We have added data on the proptosis presented by the patients during the first evaluation conducted by the radiation oncologist to the Table 4 page 6 “Baseline characteristics”. We have also included a specific mention regarding the response times to treatment in the results section line 169 page 6.

Comment 2: The proptosis is never measured. How can you say that proptosis has decreased by 3 mm ? By Hertel measurements? Comparative CT scan ?

Response 2: Thank you for pointing this out. We have specified in the 'Materials and Methods' section that patients were evaluated at baseline with CT scan and ophtalmometry; then during follow up using instrumental examinations at the discretion of the ophthalmologist and that all patients were re-evaluated with ophthalmometry during the first six months. We added more details on page 5.

Comment 3: A randomized study conducted by the MAYO clinic had shown that radiotherapy was ineffective. It is very frequent study cited. Could you precise the limitations of this study (disease duration, smoking status etc...). In brief, precise the limitations of the inclusion  criteria in randomized study

Response 3: As suggested, we have added the citation of the randomized study from the Mayo Clinic and the limitations of the study in the 'Discussion' section.

Comment 4: A maljor limitation is of course the retrospective nature of the study and the lack of control group (only steroids treated patients). This should be clearly highlighted in the limitations of the study.

Response 4: As outlined in the study design, we aimed to select patients who had undergone similar systemic treatments—specifically with corticosteroids—in order to eliminate bias related to differences in treatment efficacy. Moreover, our objective was to compare the effectiveness and safety of the same radiotherapy dose, administered using two different fractionation schemes

Comment 5: P value are lacking in table 4. It is said that the groups were comparable but it is not enough for me. The p value would be valuable. The patients are not dichotomized based on the CAS in the table. Why ? Statistics are not provided in the material section.

Response 5: Thank you for your comment. This study was designed as a retrospective observational analysis of two patient cohorts, focusing on a descriptive evaluation of clinical outcomes and symptomatic improvements following two different radiotherapy regimens. Given the nature of the study and its primary aim—to provide an initial clinical assessment rather than establish definitive statistical correlations—we have not yet conducted formal statistical analyses such as hypothesis testing with p-values or confidence interval estimations. However, the observed trends offer valuable preliminary insights that may guide further investigations. We fully recognize the importance of statistical rigor in validating our findings. As part of our ongoing research on Graves' orbitopathy, we are actively working to refine our analysis by incorporating a more comprehensive evaluation of potential confounding factors and prognostic indicators that could influence treatment response. Future studies will include appropriate statistical methodologies to strengthen our conclusions, ensuring a more robust and quantifiable assessment of therapeutic efficacy. These expanded analyses will be presented in forthcoming publications, allowing for a more detailed and statistically substantiated discussion of treatment outcomes

Comment 6: Table 5 and table 6 : who conducted these results. The ophthalkmology team ? The radiotherapy team ? How was assessed the criteria?

Response 6  We agreed and we have made a clarification about the data presented in Tables 5 and 6. Being part of a retrospective study, were extracted from the information available in the patient records during routine visits with the radiation oncologists, as defined by the study design.

Comment 7: In the conclusion, the authors state that : "Better response rate is observed in case of association with administration of systemic glucocorticoids"; this has not been demonstrated by the study. This study only showed that 10 Gy in 10 or 5 fractions were equally effective and safe.

Response 7: Thank you for your valuable feedback. We have revised the conclusions to make the objective of our study clearer and more precisely defined.

Round 2

Reviewer 3 Report

Comments and Suggestions for Authors

Thank you for your revisions